# Combination of Sarcopenia and Hypoalbuminemia Is a Poor Prognostic Factor in Surgically Treated Nonmetastatic Renal Cell Carcinoma

**DOI:** 10.3390/biomedicines11061604

**Published:** 2023-06-01

**Authors:** Tomoyuki Makino, Kouji Izumi, Hiroaki Iwamoto, Suguru Kadomoto, Atsushi Mizokami

**Affiliations:** Department of Integrative Cancer Therapy and Urology, Kanazawa University Graduate School of Medical Science, 13-1 Takara-Machi, Kanazawa 920-8640, Japan; azuizu2003@yahoo.co.jp (K.I.); hiroaki017@yahoo.co.jp (H.I.); 32f3k8@bma.biglobe.ne.jp (S.K.); mizokami@staff.kanazawa-u.ac.jp (A.M.)

**Keywords:** renal cell carcinoma, sarcopenia, albumin, psoas muscle mass index, survival, metastasis

## Abstract

Purpose: The purpose of this study is to observe how preoperative sarcopenia and hypoalbuminemia affect the oncological outcome of nonmetastatic renal cell carcinoma (RCC) after partial or radical nephrectomy. Methods: This study retrospectively analyzes 288 Japanese patients with nonmetastatic RCC who underwent radical treatment at Kanazawa University Hospital between October 2007 and December 2018. Relationships between sarcopenia as indicated by the psoas muscle mass index and hypoalbuminemia (albumin ≤ 3.5 g/dL) with overall survival (OS) and metastasis-free survival (MFS) were determined. Results: The study found that 110 (38.2%) of the 288 patients were sarcopenic and 29 (10.1%) had hypoalbuminemia. The combination of sarcopenia and hypoalbuminemia was associated with a shorter OS and MFS (*p* for trend = 0.0007 and <0.0001, respectively), according to Kaplan–Meier curves. The concurrent presence of sarcopenia and hypoalbuminemia were found to be significant and independent predictors of poor MFS (hazard ratio (HR), 2.96; 95% confidence interval (95% CI), 1.05–8.39; *p* = 0.041) and poor OS (HR, 6.87; 95% CI, 1.75–26.94; *p* = 0.006), respectively. Conclusions: In Japanese patients with surgically treated nonmetastatic RCC, combined preoperative sarcopenia and hypoalbuminemia was a significant predictor of poor survival.

## 1. Introduction

Predictive tools to accurately assess the risk of mortality, recurrence, and postoperative complications are extremely useful in real-world clinical practice. At present, prognostic models, such as pathological tumor stages, tumor size, nuclear grade, and histologic tumor necrosis, are tumor-centric and require data derived from the final pathology specimen [1,2]. However, cancer prognosis is influenced not only by tumor characteristics, but also by a patient’s physical condition, with a particular emphasis on inflammatory and nutritional status. The development and validation of prognostic models that incorporate readily available baseline clinical data to inform decision making and treatment selection are gaining popularity.

Sarcopenia is a progressive and common skeletal muscle disease characterized by reduced skeletal muscle mass and function [3]. Sarcopenia pathogenesis is complex, and several mechanisms are thought to contribute to the phenotype, including hormone function, redistribution of muscle fiber, decreased number of motor units, decreased number and regenerative capacity of satellite cells, inflammatory pathways, and intracellular changes in proteostasis and mitochondrial function [4]. Several existing studies suggest that skeletal muscle mass more accurately represents body composition than body mass index (BMI), demonstrating that BMI is an inadequate surrogate for both muscle mass and adiposity, and is not consistently associated with survival in patients with several cancers [5,6]. In contrast, several recent studies have confirmed that the onset of sarcopenia is closely related to the treatment and prognosis of many cancers, including hepatocellular, pancreaticobiliary, gastroesophageal, colorectal, and urothelial carcinomas [7,8].

Serum albumin is a simple and inexpensive marker of systemic inflammation, as well as a surrogate for nutritional status and a strong prognostic factor in cancer patients [9,10]. Furthermore, albumin has been shown to be a prognostic factor and predictive marker of treatment response in many cancer types when combined with C-reactive protein (CRP), fibrinogen, globulin, and lymphocyte [11,12,13,14,15,16,17,18]. Malnutrition, routinely indicated by decreased in serum albumin and low BMI, is often comorbid with sarcopenia and may accelerate the process of muscle degeneration [19]. Although the independent association of sarcopenia and albumin with mortality in patients with renal cell carcinoma (RCC) is well known, there have been few studies on the relationship between sarcopenia and albumin status and RCC. Thus, this study retrospectively aims to investigate the potential associations between sarcopenia and hypoalbuminemia and oncological outcomes in Japanese patients with nonmetastatic RCC after curative surgery.

## 2. Patients and Methods

### 2.1. Patients

This retrospective study included patients with nonmetastatic RCC (T1–T4, N0, and M0), who underwent partial or radical nephrectomy at Kanazawa University Hospital between October 2007 and December 2018. According to our previous study [14], the inclusion and exclusion criteria were established. The inclusion criteria were set as follows: (I) age of 18 years or older; (II) confirmed imaging or histologic diagnosis of RCC; and (III) complete electronic medical records, including clinical laboratory tests within one month before surgery. We excluded patients who did not undergo surgical therapy. Furthermore, the Medical Ethics Committee of Kanazawa University (2018-116) approved this study [14]. All research was performed in accordance with relevant guidelines and regulations and the Declaration of Helsinki. The requirement for informed consent was waived by the Medical Ethics Committee of Kanazawa University due to the observational nature of the study using only existing data. Instead, information on this study will be posted on the Kanazawa University Hospital website, and patients are free to revoke their consent at any point.

### 2.2. Data Collection and Variable Definitions

Age, gender, BMI, psoas muscle mass index (PMI), Karnofsky performance status (KPS), Charlson comorbidity index (CCI), smoking history, medical conditions, such as hypertension and diabetes, and preoperative serum biomarkers, such as CRP and albumin concentration, were collected at the time of surgery. In addition, baseline oncological data were obtained, including pathological tumor stage and size, histological subtype, histological nuclear grade, and lymphovascular invasion. The pathological stage was determined using the Union for International Cancer Control’s (2017) tumor–node–metastasis (TNM) classification of malignant tumors (eighth edition).

Overall survival (OS) and metastasis-free survival (MFS) were calculated as the time from the date of surgery to death from any cause and the first detection of RCC metastasis, respectively.

### 2.3. Assessment of Sarcopenia and Hypoalbuminemia

The PMI is a straightforward method of expressing total body skeletal muscle mass that is often used to assess sarcopenia cases. The total psoas muscle cross-sectional area at the L3 vertebral level was manually measured by tracing preoperative computed tomography images, and PMI (mm^2^/m^2^) was calculated by multiplying the total psoas muscle area (mm^2^) by the square of the patient’s height (m^2^) [20]. In this study, the optimal cutoff value for PMI was determined by using the point closest to (0, 1) on the receiver operating curve (ROC) [21]. A 3.5 g/dL cutoff was used to distinguish between normal and low serum albumin levels according to a previous study [22]. Patients were divided into four groups to study the effects of sarcopenia and serum albumin together: non-sarcopenia and albumin > 3.5, sarcopenia and albumin > 3.5, non-sarcopenia and albumin ≤ 3.5, and sarcopenia and albumin ≤ 3.5.

### 2.4. Statistical Analysis

The chi-squared test and one-way analysis of variance (ANOVA) were used to compare differences in the patients’ characteristics. The Kaplan–Meier method was used to estimate OS and MFS, and the log-rank and log-rank trend tests were used to compare them. To assess the association of sarcopenia and albumin status with MFS and OS, univariate and multivariate analyses were performed using Cox proportional hazards models. Statistical analyses were performed using GraphPad Prism version 6.07 (GraphPad Software Inc., San Diego, CA, USA) and IBM SPSS Statistics version 25 (IBM Corp., Armonk, NY, USA). Statistical significance was indicated by a *p*-value of <0.05.

## 3. Results

### 3.1. Patient and Disease Characteristics

The data were extracted from 288 patients with T1–T4, N0, and M0 RCC who underwent partial or radical nephrectomies. Table 1 shows the demographics of all the patients. The study population’s median follow-up period was 4.38 yr (range, 0.02–14.71 yr). Most patients were males (72.2%) and the median age of the 288 patients was 63 yr (interquartile range (IQR), 55–71 yr), and 110 (38.2%) and 29 (10.1%) patients were diagnosed with sarcopenia and hypoalbuminemia (albumin ≤ 3.5 g/dL), respectively. The median BMI, PMI, serum albumin concentration, serum CRP level, and tumor size of the entire cohort were 23.7 kg/m^2^ (IQR, 21.7–26.4 kg/m^2^), 468.2 mm^2^/m^2^ (IQR, 352.9–571.7 mm^2^/m^2^), 4.2 g/dL (IQR, 3.9–4.5 g/dL), 0.1 mg/dL (IQR, 0.1–0.2 mg/dL), and 2.9 cm (IQR, 2.0–4.93 cm), respectively. Patients had a radical nephrectomy (50.3%), clear cell histology (85.1%), ≥pT3 (17.7%), nuclear grade ≥ 3 disease (17.0%), and lymphovascular invasion (47.9%).

### 3.2. The Optimal Thresholds for PMI

The ROC analysis was used to determine the optimal cutoff value for PMI using OS as the end point. The ROC revealed that the optimal PMI cutoff value for males was 516.8 mm^2^/m^2^ (area under curve (AUC), 0.573; 95% CI, 0.457–0.688; *p* = 0.3081, with sensitivity of 61.1% and specificity of 53.8%) and 235.1 mm^2^/m^2^ for females (AUC, 0.686; 95% CI, 0.437–0.934; *p* = 0.2118, with sensitivity of 50%). Sarcopenia was defined as any measurement that was lower than each PMI value.

### 3.3. Association of Clinicopathological Parameters with Sarcopenia and Albumin

As shown in Table 2, patients were stratified into groups based on their sarcopenia and albumin status to investigate significant associations between covariates using ANOVA. The variables of age, gender, KPS, CCI, BMI, CRP, tumor size, pathological T-stage, nuclear grade, and lymphovascular invasion demonstrated significant associations within an integrated model considering both sarcopenia and hypoalbuminemia.

### 3.4. Survival Rates and Prognostic Factors

During the study period, 42 patients developed metastatic recurrence and 21 patients died, eight of whom died from RCC. Figure 1 shows the results of the Kaplan–Meier analysis for OS and MFS in all patients stratified by sarcopenia and albumin status. There was a statistically significant difference in OS (*p* = 0.0011), and concurrent sarcopenia and hypoalbuminemia tended to worsen survival (*p* for trend = 0.0007, Figure 1a). The 5-year OS values for non-sarcopenia and albumin > 3.5, sarcopenia and albumin > 3.5, non-sarcopenia and albumin ≤ 3.5, and sarcopenia and albumin ≤ 3.5 groups were 96.4%, 93.7%, 82.5%, and 86.5%, respectively. Furthermore, the MFS varied significantly (*p* < 0.0001) between the four groups, and concurrent sarcopenia and hypoalbuminemia tended to increase metastasis (*p* for trend < 0.0001, Figure 1b). The 5-year MFS values for non-sarcopenia and albumin > 3.5, sarcopenia and albumin > 3.5, non-sarcopenia and albumin ≤ 3.5, and sarcopenia and albumin ≤ 3.5 groups were 90.9%, 81.3%, 58.3%, and 42.9%, respectively. Sarcopenia and albumin status were significantly associated with shorter MFS in the multivariate analysis of the prognostic factors (non-sarcopenia, albumin ≤ 3.5: hazard ratio (HR), 4.79; 95% CI, 1.44–15.89; *p* = 0.010; sarcopenia, albumin ≤ 3.5: HR, 2.96; 95% CI, 1.05–8.39; *p* = 0.041, Table 3). Furthermore, for the multivariate analysis, a stepwise increase in HRs and a decrease in *p*-values were observed with increasing risk in the OS composite prognostic models that included both sarcopenia and hypoalbuminemia (sarcopenia, albumin > 3.5: HR, 3.64; 95% CI, 1.21–10.90; *p* = 0.021; non-sarcopenia, albumin ≤ 3.5: HR, 5.87; 95% CI, 1.39–24.73; *p* = 0.016; sarcopenia, albumin ≤ 3.5: HR, 6.87; 95% CI, 1.75–26.94; *p* = 0.006, Table 4).

## 4. Discussion

The need for better prognostic models based on preoperative parameters in localized RCC remains an important challenge. Previous research has demonstrated a significant association between sarcopenia and an elevated risk of mortality and recurrence following nephrectomy in both localized and metastatic RCCs [23,24,25,26,27]. Furthermore, a recent meta-analysis found that patients with malignant neoplasms, including RCC and sarcopenia, had worse clinical outcomes than those who did not [28]. Interestingly, recent research has shown that preoperative sarcopenia and elevated systemic inflammatory markers, such as the modified Glasgow Prognostic Score including both CRP and albumin levels, were associated with a decreased survival [29,30]. Furthermore, a recent retrospective study found that patients with localized RCC who had both sarcopenia and hypoalbuminemia prior to surgery had a two-to-three-fold reduction in OS and recurrence-free survival after nephrectomy in an American population [22]. Hypoalbuminemia, weight loss, and low BMI are commonly utilized indicators for evaluating nutritional deficiencies in patients with RCC [31]. Nevertheless, relying solely on BMI to elucidate changes in body mass may yield unreliable prognostic information, as it may not accurately capture specific shifts between lean and adipose tissues [32]. This is particularly significant in the case of patients with sarcopenic obesity, as the presence of stable or increased adiposity may camouflage detrimental alterations in skeletal muscle [6]. In contrast, sarcopenia has been consistently linked to nutritional deficiencies in both cancer and non-cancer patients [3]. Overall, assessing body composition and nutritional status prior to surgery may help urologists and oncologists make treatment decisions.

The present study examined the relationship between preoperative sarcopenia and albumin status, as well as postoperative survival outcomes in Japanese patients with nonmetastatic RCC who had nephrectomy. The findings confirm that the presence of sarcopenia, as indicated by low PMI, and hypoalbuminemia is not only associated with significantly worse MFS values, but also with decreased OS after radical surgery for nonmetastatic RCC, and that a combined model of sarcopenia and hypoalbuminemia is an independent poor prognostic factor for MFS and OS.

The correlation between sarcopenia and heightened mortality could potentially be attributed to the interplay between factors, such as low muscle mass, malnutrition, and systemic inflammation [22]. Accumulating evidence indicates that the nutritional and immune statuses are involved in the onset and advancement of cancer, consequently impacting survival outcomes [33]. Sarcopenia was found to be an independent predictor of OS (HR, 1.83; 95% CI, 1.41–2.37), cancer-specific survival (CSS) (HR, 1.78; 95% CI, 1.34–2.36), and progression-free survival (PFS) (HR, 1.98; 95% CI, 1.34–2.92) in a recent meta-analysis involving 3591 patients with RCC [34]. Furthermore, a recent umbrella review concluded that sarcopenia was significantly associated with multiple health-related outcomes, such as dysphagia, cognitive impairment, fractures, falls, and hospitalization in older populations, whether or not tumors were present [35]. Otherwise, in a prior investigation conducted at our institution, which encompassed 299 patients with surgically treated nonmetastatic RCC, we presented evidence indicating that sarcopenia served as a substantial indicator of unfavorable pathological outcomes and diminished survival prognosis [27]. Serum albumin, on the other hand, is an objective indicator of nutritional status and clinical inflammation, implying that its expression is reduced in inflammatory conditions [36]. Hypoalbuminemia has been linked to an increased in overall mortality in patients undergoing surgery for both localized and metastatic RCCs [31,37]. Furthermore, a recent meta-analysis involving 2863 patients with metastatic RCC treated with tyrosine kinase inhibitors showed that a lower pre-treatment serum albumin level was related to poorer OS (dichotomous: HR, 2.01; 95% CI, 1.64–2.46; *p* < 0.001; *I*^2^ = 28.8%; continuous: HR, 0.93; 95% CI, 0.86–1.00; *p* = 0.040; *I*^2^ = 67.5%) [38]. Interestingly, recent oncologic studies evaluating the utility of serum albumin in prognostic models have primarily evaluated it in combination with other biomarkers. The prognostic nutritional index (PNI), which integrates lymphocyte count and serum albumin levels, is widely regarded as a comprehensive measure reflecting the nutritional and immunological status related to cancer. As patients with a lower PNI may experience a compromised antitumor response and subsequently exhibit decreased survival rates, PNI serves as a valuable tool for clinicians to anticipate optimal preoperative medical interventions and determine the ideal timing for surgical procedures [33,39]. A decreased PNI was found to be a significant predictor of worse OS (HR, 2.00; 95% CI, 1.64–2.42; *p* < 0.001), CSS (HR, 2.54; 95% CI, 1.61–4.00; *p* < 0.001), and PFS (HR, 2.12; 95% CI, 1.82–2.46; *p* < 0.001) in a recent meta-analysis involving 7629 patients with RCC [40]. Otherwise, in a previous study at our institution involving 213 patients with nonmetastatic RCCs, we demonstrated that the combination of serum CRP and albumin is a predictor of postoperative recurrence [14].

Because sarcopenia and hypoalbuminemia strongly reflect nutritional status and systemic inflammation and are independently associated with poor treatment outcomes and survival in RCC, it stands to reason that this combined model can be used as a powerful prognostic marker in patients with nonmetastatic RCC. To our knowledge, this study represents the initial investigation into the biomarkers of sarcopenia and hypoalbuminemia, specifically in Japanese patients with nonmetastatic RCC. All the biomarkers assessed in this study are readily obtainable through routine blood tests or imaging studies, offering the potential for straightforward and replicable prognostic markers, distinct from the previous research. Thus, the pretreatment assessment of sarcopenia and measurement of albumin levels may be of great help in predicting prognosis after surgical treatment for patients with nonmetastatic RCC. Based on these findings, clinicians can contribute to the identification of patients with a compromised nutritional status and poor body composition who may potentially benefit from additional resources and interventions.

Early intervention with muscle-building exercise programs and nutritional supplements linked to both myoprotein synthesis and anti-inflammatory pathways has been shown to reduce mortality and improve muscle composition in cancer patients [29,41,42]. Interestingly, a recent study discovered that the dynamics in sarcopenia status from pre- to postoperative stages served as a substantial prognostic factor for survival outcomes, suggesting the significance of maintaining optimal nutritional status before and after surgery to enhance long-term survival in patients with RCC [43]. Nutritional interventions aim to maintain or improve dietary intake, skeletal muscle mass, and physical performance, and reduce metabolic abnormalities. To meet all nutrient and micronutrient requirements, cancer patients should consume a high-energy diet rich in high-density foods, including fats, and have a high-protein intake. Furthermore, artificial nutrition and antioxidant supplementation should be used as needed. Multimodal training (combining various methods of physical training, such as aerobic and strength exercises, with pharmacotherapy and nutritional supplementation) can be an interesting strategy for improving the results [44]. Thus, conducting preoperative assessments of body composition and nutritional status to identify patients with modifiable risk factors, along with implementing a multimodal approach to managing body composition throughout all stages of the disease, may play a crucial role in enhancing patient outcomes.

There were several limitations to this study. Manual tracing was used to collect PMI data; however, this study did not investigate the correlation between PMI and total skeletal muscle mass. Furthermore, it should be noted that the cutoff values for PMI varied among different racial populations, and specific cutoff values for each race were not established. Furthermore, because of the complex sequential treatment using various currently available agents, such as molecularly targeted drugs and immune checkpoint inhibitors, the systemic treatment of patients with recurrent metastatic RCC after radical surgery was not detailed in this study. Furthermore, recent significant advances in surgical treatment may have had an impact on these prognostic analyses. Finally, the sample size and observation period may be insufficient to precisely determine the statistical significance. The present study, on the other hand, confirmed that sarcopenia, defined here as low PMI, and hypoalbuminemia are both independent predictors of survival in patients surgically treated for nonmetastatic RCC. More large-scale studies are needed to confirm these findings.

## 5. Conclusions

According to the findings of this retrospective study, Japanese patients with nonmetastatic RCC and concurrent sarcopenia and hypoalbuminemia had a higher incidence of metastasis and poor prognosis. Evaluating body composition and albumin status may be useful for prognostic risk stratification in patients with nonmetastatic RCC who are undergoing surgical treatment, as well as for identifying patients who require early nutritional and exercise interventions aiming to maintain or improve food intake and skeletal muscle mass and function.

## Figures and Tables

**Figure 1 biomedicines-11-01604-f001:**
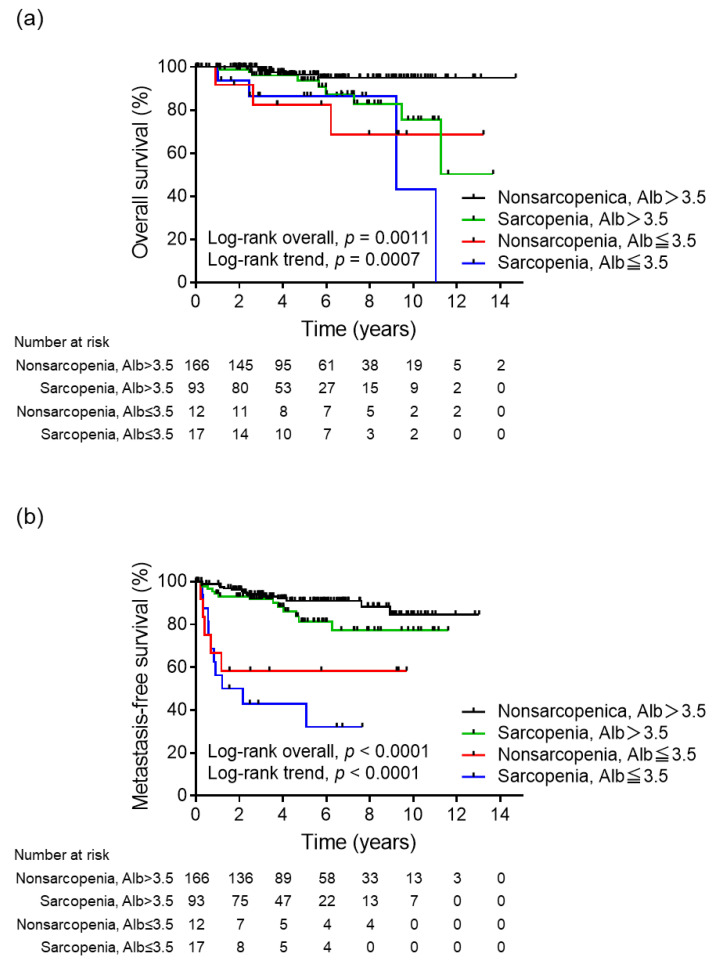
Kaplan–Meier analyses for (**a**) overall survival and (**b**) metastasis-free survival based on the sarcopenia and albumin status, respectively.

**Table 1 biomedicines-11-01604-t001:** Patient demographics (*n* = 288).

Characteristics	*n* (%)
Median age, yr (IQR)	63 (55–71)
Sex	
Male	208 (72.2)
Female	80 (27.8)
Median BMI, kg/m^2^ (IQR)	23.7 (21.7–26.4)
BMI ≥ 25 kg/m^2^	100 (34.7)
Median PMI, mm^2^/m^2^ (IQR)	468.2 (352.9–571.7)
Male	520.8 (440.5–619.0)
Female	299.2 (249.2–374.8)
Sarcopenia	110 (38.2)
KPS ≤ 90	17 (5.9)
Median CCI (IQR)	2.0 (1.0–3.0)
CCI ≥ 3	121 (42.0)
Hypertension	153 (53.1)
Diabetes	54 (18.8)
Smoking status	155 (53.8)
Median albumin concentration, g/dL (IQR)	4.2 (3.9–4.5)
Hypoalbuminemia (≤3.5 g/dL)	29 (10.1)
Median CRP level, mg/dL (IQR)	0.1 (0.1–0.2)
Elevated CRP (≥0.3 mg/dL)	62 (21.5)
Sarcopenia and albumin status	
Non-sarcopenia, albumin > 3.5	166 (57.6)
Sarcopenia, albumin > 3.5	93 (32.3)
Non-sarcopenia, albumin ≤ 3.5	12 (4.2)
Sarcopenia, albumin ≤ 3.5	17 (5.9)
Surgery type	
Radical nephrectomy	145 (50.3)
Partial nephrectomy	143 (49.7)
Histology	
Clear cell	245 (85.1)
Non-clear cell	43 (14.9)
Median tumor size, cm (IQR)	2.9 (2.0–4.93)
Pathological T-stage	
≤pT2	237 (82.3)
≥pT3	51 (17.7)
Grade	
≤G2	226 (78.5)
≥G3	49 (17.0)
Unknown	13 (4.5)
Lymphovascular invasion	
Yes	138 (47.9)
No	148 (51.4)
Unknown	2 (0.7)

Abbreviations: BMI, body mass index; CCI, Charlson comorbidity index; CRP, C-reactive protein; IQR, interquartile range; KPS, Karnofsky performance status; PMI, psoas muscle mass index.

**Table 2 biomedicines-11-01604-t002:** Summary of clinicopathological characteristics according to preoperative sarcopenia and albumin status.

		Sarcopenia and Albumin Status				
Covariate	Level	Non-Sarcopenia, Albumin > 3.5 (*n* = 166) (%)	Sarcopenia, Albumin > 3.5 (*n* = 93) (%)	Non-Sarcopenia, Albumin ≤ 3.5 (*n* = 12) (%)	Sarcopenia, Albumin ≤ 3.5 (*n* = 17) (%)	*p*-Value
Age	≥65	60 (36.1)	58 (62.4)	5 (41.7)	9 (52.9)	<0.001
Sex	Male	103 (62.0)	84 (90.3)	6 (50.0)	15 (88.2)	<0.001
KPS	≤90	6 (3.6)	3 (3.2)	4 (33.3)	4 (23.5)	<0.001
CCI	Median (IQR)	2.0 (1.0–3.0)	3.0 (2.0–4.0)	2.0 (0.8–3.3)	3.0 (2.0–3.0)	0.009
	≥3	57 (34.3)	51 (54.8)	4 (33.3)	9 (52.9)	0.002
BMI (kg/m^2^)	Median (IQR)	24.3 (22.5–26.8)	23.1 (21.4–24.9)	25.8 (22.4–27.6)	22.3 (20.5–23.3)	<0.001
	≥25	67 (40.4)	23 (24.7)	7 (58.3)	3 (17.6)	0.009
CRP level (mg/dL)	Median (IQR)	0.10 (0–0.20)	0.10 (0.10–0.20)	0.80 (0.10–5.33)	3.00 (0.20–8.70)	<0.001
	≥0.3	22 (13.3)	21 (22.6)	7 (58.3)	12 (70.6)	<0.001
Tumor size (cm)	Median (IQR)	2.7 (1.9–4.48)	2.9 (2.0–4.50)	5.05 (2.48–10.63)	6.6 (2.5–8.8)	<0.001
	>4	44 (26.5)	28 (30.1)	8 (66.7)	11 (64.7)	<0.001
Pathological T-stage	≤pT2	147 (88.6)	73 (78.5)	7 (58.3)	10 (58.8)	<0.001
	≥pT3	19 (11.4)	20 (21.5)	5 (41.7)	7 (41.2)	
Grade	≤G2	140 (84.3)	74 (79.6)	7 (58.3)	5 (29.4)	<0.001
	≥G3	16 (9.6)	16 (17.2)	5 (41.7)	12 (70.6)	
	Unknown	10 (6.0)	3 (3.2)	0	0	
Lymphovascular invasion	Yes	73 (44.0)	47 (50.5)	5 (41.7)	13 (76.5)	0.072
	No	92 (55.4)	45 (48.4)	7 (58.3)	4 (23.5)	
	Unknown	1 (0.6)	1 (1.1)	0	0	

Abbreviations: BMI, body mass index; CCI, Charlson comorbidity index; CRP, C-reactive protein; IQR, interquartile range; KPS, Karnofsky performance status.

**Table 3 biomedicines-11-01604-t003:** Univariate and multivariate analyses of the association between clinicopathological characteristics and metastasis-free survival.

Variables		Univariate	Multivariate	
		*p*-Value	*p*-Value	HR (95% CI)
Age	≥65 vs. <65	0.053	0.017	0.43 (0.22–0.86)
Sex	Male vs. female	0.151		
KPS	≤90 vs. 100	<0.001	0.047	2.46 (1.01–5.99)
CCI	≥3 vs. <3	0.350		
BMI	≥25 vs. <25	0.486		
Hypertension	Yes vs. no	0.655		
Diabetes	Yes vs. no	0.602		
Smoking	Yes vs. no	0.943		
CRP level	≥0.3 vs. <0.3	<0.001	0.810	1.12 (0.45–2.80)
Tumor size	>4 cm vs. ≤4 cm	<0.001	0.001	4.79 (1.88–12.20)
Pathological T-stage	3–4 vs. 1–2	<0.001	0.835	1.10 (0.46–2.61)
Histologic type	Non-ccRCC vs. ccRCC	0.485		
Grade	≥3 vs. <3	<0.001	0.003	3.26 (1.48–7.17)
Lymphovascular invasion	Yes vs. no	<0.001	0.083	2.31 (0.90–5.97)
Sarcopenia + albumin status	Non-sarcopenia, albumin > 3.5	—	—	1 (ref.)
	Sarcopenia, albumin > 3.5	0.124	0.102	1.94 (0.88–4.32)
	Non-sarcopenia, albumin ≤ 3.5	<0.001	0.010	4.79 (1.44–15.89)
	Sarcopenia, albumin ≤ 3.5	<0.001	0.041	2.96 (1.05–8.39)

Abbreviations: BMI, body mass index; CCI, Charlson comorbidity index; ccRCC, clear cell renal cell carcinoma; CI, confidential interval; CRP, C-reactive protein; HR, hazard ratio; KPS, Karnofsky performance status.

**Table 4 biomedicines-11-01604-t004:** Univariate and multivariate analyses of the association between clinicopathological characteristics and overall survival.

Variables		Univariate	Multivariate	
		*p*-Value	*p*-Value	HR (95% CI)
Age	≥65 vs. <65	0.314		
Sex	Male vs. female	0.302		
KPS	≤90 vs. 100	0.082	0.116	2.77 (0.78–9.84)
CCI	≥3 vs. <3	0.584		
BMI	≥25 vs. <25	0.717		
Hypertension	Yes vs. no	0.271		
Diabetes	Yes vs. no	0.610		
Smoking	Yes vs. no	0.276		
CRP level	≥0.3 vs. <0.3	0.708		
Tumor size	>4 cm vs. ≤4 cm	0.276		
Pathological T-stage	3–4 vs. 1–2	0.366		
Histologic type	Non-ccRCC vs. ccRCC	0.602		
Grade	≥3 vs. <3	0.450		
Lymphovascular invasion	Yes vs. no	0.082	0.238	1.74 (0.69–4.35)
Sarcopenia + albumin status	Non-sarcopenia, albumin > 3.5	—	—	1 (ref.)
	Sarcopenia, albumin > 3.5	0.022	0.021	3.64 (1.21–10.90)
	Non-sarcopenia, albumin ≤ 3.5	0.012	0.016	5.87 (1.39–24.73)
	Sarcopenia, albumin ≤ 3.5	0.001	0.006	6.87 (1.75–26.94)

Abbreviations: BMI, body mass index; CCI, Charlson comorbidity index; ccRCC, clear cell renal cell carcinoma; CI, confidential interval; CRP, C-reactive protein; HR, hazard ratio; KPS, Karnofsky performance status.

## Data Availability

The data presented in this study are available on request from the corresponding author.

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
