# Peer review of "Combination of Sarcopenia and Hypoalbuminemia Is a Poor Prognostic Factor in Surgically Treated Nonmetastatic Renal Cell Carcinoma"

_biomedicines, 2023, doi:10.3390/biomedicines11061604_

Round 1

Reviewer 1 Report

The authors present the outcomes from a retrospective series of Japanese patients with the aim of investigating the potential associations between sarcopenia and hypoalbuminemi a and oncological outcomes in patients with nonmetastatic RCC after curative surgery.

The title reflects the subject of the manuscript. It presents a clear and clinically useful message. It is well written in terms of clarity, style, and use of English language. The discussion section is sufficiently detailed and explains adequately the purpose of this study in the context of published information. The conclusion accurately and clearly explains the main result. The length of the manuscript is ideal. All references are appropriate and current.

Minor points: 

- I would suggest adding R status on the patient characteristics table

Author Response

Thank you very much for insightful suggestions. Does “R status” mean surgical margin status? Unfortunately, this proposal could not be fulfilled in this study because we were not able to follow up all patients for evaluation of resection margins.

Reviewer 2 Report

This study is interesting, but there are some points to revise before publishing. 1 How did you decide the cutoff value of albumin? Could you show reference? 2 Authors said that "Although the independent 55 association of sarcopenia and albumin with mortality in patients with renal cell carcinoma (RCC) is well known", and the albumin >3.5 is more strongly affected the clinical outcomes in your study, I thought. Is there any advantage of using both albumin >3.5 and sarcopenia?   3 I think comorbidity may influence both sarcopenia and albumin. Could you tell us Charlson Comorbidity Index?

There is some points to revise before publishing.

Author Response

1 How did you decide the cutoff value of albumin? Could you show reference?

Thank you very much for mentioning to important point. We set the cutoff value of albumin according to previous studies (revised page 3, line 97, reference No. 22; Int J Clin Oncol 2019;24(6):698–705; and J Oncol 2018:1953571). Moreover, albumin 3.5 g/dL or less is called "low nutrition” in Japan. Therefore, a 3.5 g/dL cutoff was used to distinguish between normal and low serum albumin levels.

2 Authors said that "Although the independent 55 association of sarcopenia and albumin with mortality in patients with renal cell carcinoma (RCC) is well known", and the albumin >3.5 is more strongly affected the clinical outcomes in your study, I thought. Is there any advantage of using both albumin >3.5 and sarcopenia?

Thank you very much for important suggestions. We believe that the combination of sarcopenia and hypoalbuminemia is additive impact in shortening overall survival (OS) and metastasis-free survival (MFS). According to the results of Kaplan–Meier analysis, significant differences were observed in OS between non-sarcopenia/albumin >3.5 and non-sarcopenia/albumin ≤3.5 (p = 0.0015) and sarcopenia/albumin >3.5 (p = 0.0139), respectively. Further, although there was no significant difference in MFS between non-sarcopenia/albumin >3.5 and sarcopenia/albumin >3.5 (p = 0.1130), significant difference was observed between non-sarcopenia/albumin >3.5 and non-sarcopenia/albumin ≤3.5 (p < 0.0001). Thus, a combined model of sarcopenia and hypoalbuminemia may be a more powerful prognostic model than a single model of each.

3 I think comorbidity may influence both sarcopenia and albumin. Could you tell us Charlson Comorbidity Index?

Thank you very much for insightful comments. According to your suggestions, the Charlson Comorbidity Index (CCI) was added to the patient characteristics tables (revised page in 3, Table 1; and revised page in 5, Table 2) and reanalyzed univariate and multivariate analyses (revised page in 6, Table 3 and 4). As a result, CCI had no impact on prognosis.